# Plastic film residues on cropland: monitoring soil contamination through optical remote sensing

Alessandro Fabrizi<sup>1</sup>, Peter Fiener<sup>1</sup>, Kristof Van Oost<sup>2</sup>, Florian Wilken<sup>1</sup>

- <sup>1</sup> Institute of Geography, University of Augsburg, Augsburg, Germany
- <sup>2</sup> Earth and Life Institute, Université Catholique de Louvain, Louvain-la-Neuve, Belgium

Correspondence to: Florian Wilken (florian.wilken@uni-a.de)

**Abstract.** Plastic films have been improving agricultural production and covering an increasing surface area of cropland in the last decades. Yet their use has been connected to the generation of plastic residues, potentially acting as a main secondary microplastic source in agricultural soils. Monitoring the generation of plastic film residues is crucial for identifying good management practices and assessing the risk of plastic use in agriculture. Remote sensing has been qualified as a valuable tool for monitoring macroplastic mainly on waters, while its use on agricultural soils is mostly unexplored. Our study combined proximal and remote sensing techniques to lay the foundations of UAV (Unmanned Aerial Vehicle) use for monitoring macroplastic film residues on cropland.

Through proximal and UAV acquisitions of five-bands multispectral data (i.e., blue, green, red, red edge, near infrared), we highlighted the potential of off-the-shelf miniaturised sensors and identified possible workflows for detecting macroplastic film residues. Our findings highlight a greater efficacy of spatial resolution over spectral resolution, encouraging the use of high-resolution RGB cameras over multispectral cameras. Through proximal acquisitions of hyperspectral data, we built spectral libraries and located absorption peaks for the most commonly used plastic films. We highlighted that these absorption peaks unambiguously identify plastic films on cropland and offer the potential to distinguish plastic types, encouraging the development of sensors tailored for plastic detection.

#### 1. Introduction

20

Soils are estimated to be a greater plastic sink than waters (Kedzierski et al., 2023; Lofty et al., 2022) and to potentially act as a plastic source for other environmental matrices through water and wind erosion (Rehm et al., 2021; Rezaei et al., 2022). In experimental setups high to very high concentrations of plastic have been shown to degrade soil physicochemical properties and affect plant health (Zhang et al., 2022), but the threshold concentrations at which these effects occur are still poorly quantified. Most likely, the source and physiochemical properties of the plastic are important drivers (Wang et al., 2023b; Wang et al., 2023a; Liu et al., 2023), highlighting the necessity of tracing the origin of plastic contamination. Despite the scarce knowledge about the

toxicity to biota of current plastic exposure levels, the ubiquitous presence, persistence, bioaccumulation, and possible unknown effects of plastic threaten soil health and call for plastic contamination monitoring plans.

Plastic in soil may span a wide range of polymers, shapes, sizes, and sources. One of the major sources and sinks of plastic are agricultural soils (Horton et al., 2017; Kawecki and Nowack, 2019), since plastic is extensively used in various ways to enhance productivity such as mulching and greenhouse films, or seed and fertiliser coatings (Fao, 2021; Eip-Agri Focus Group, 2021). Around 80 % of plastic use in European vegetable production is attributed to plastic films used as crop covers (Agriculture Plastic Environment Europe, 2019). While plastic films have been associated with the generation of macro- and microplastic residues (> 5 mm and 

65

absorption features and hyperspectral imaging for the ex situ recognition of macroplastic residues found on agricultural soils. Still, reference spectral libraries are currently not available for agricultural plastics, and their establishment has been shown to push forward remote sensing algorithms for plastic detection (Garaba et al., 2018; Garaba et al., 2021). Despite the high potential of SWIR-based plastic detection, it must be considered that the absorption bands are typically available on hyperspectral sensors only, and their limited availability may hamper large-scale monitoring. Multispectral cameras offer a good compromise between data volume and spectral resolution, and the availability of off-the-shelf miniaturised sensors has favoured their use on low-payload UAVs. To the best of our knowledge, multispectral cameras and the non-visible part of the spectrum in general have not been explored for remote sensing detection of plastics residues on agricultural soils. RGB cameras further decrease the complexity of data collection and analysis, and they increase spatial resolution at the cost of spectral resolution. Their use, coupled with deep learning, has been shown to result in reasonable plastic film residues mapping on cropland, but it has been limited to the detection of transparent film residues on cotton fields (Zhai et al., 2022; Qiu et al., 2022; Yang et al., 2024). Different film and soil types may require seeing beyond the visible.

Our work aimed at a comprehensive assessment of UAV optical remote sensing use to detect different plastic film residues on cropland. We combined hyperspectral and multispectral proximal sensing with multispectral remote sensing to (i) build spectral libraries for the most common plastic films used in agriculture; (ii) define spectral changes occurring when moving from pristine plastic film to residue in soil; (ii) define possible workflows and sensors for detecting different plastic film residues using UAV technology.

# 2. Methods

# 2.1 Proximal sensing

## 2.1.1 Experimental site and measuring devices

Outdoor measurements of plastic film reflectance were carried out on a rooftop in Louvain-la-Neuve, Belgium, between 10:00 and 18:00 on 24 July 2022, with almost clear sky conditions. The measurements were performed through the parallel use of a spectroradiometer (ASD fieldspec 3; Fig. 1a; Malvern Panalytical, Ltd.), operated via an acquisition software (RS<sup>3</sup>; Malvern Panalytical, Ltd.), and a multispectral camera (Micasense RedEdge-MX RX02; Fig. 1b; AgEagle Aerial Systems, Inc.), using a setup similar to Crucil et al. (2019).

The spectroradiometer was used with a fiberoptic cable mounted on a pistol grip (Fig. 1a) to acquire hyperspectral data with 3 nm spectral resolution in the 350–1000 nm region, and 10 nm spectral resolution in the 1000–2500 nm region. The software automatically performed the conversion to reflectance measurements and output data at a 1 nm interval. A Spectralon panel (Fig. 1a) was used as a white reference to calibrate the

105

instrument about every 4 minutes, and all the spectra were obtained from the average of 75 spectra, each with 34 ms of integration time. The fiberoptic tip was placed 11 cm above the targets at the nadir position, providing a conical field of view with a diameter of 4.9 cm.

The multispectral camera used five imaging sensors to acquire spectral data centred at 475, 560, 668, 717, 842 nm wavelengths – namely blue, green, red, red edge, and NIR (near infrared) bands – with bandwidths of 32, 27, 16, 12, 57 nm, respectively. Images of a reflectance panel were acquired before every measurement and used to calibrate and convert data to reflectance values in a later stage. The camera was placed 70 cm above the targets at the nadir position, providing around 0.5 mm/pixel images.

Figure 1 – Experimental setup used for acquiring reflectance measurements of plastic films from a spectroradiometer (i.e., ASD) (a) and a multispectral camera (b). The same batch of samples is represented on the left (a) and on the right part of the image (b). On the left side (a), rows and columns were labelled to show different samples and treatments: rows are labelled according to the film type, following the abbreviation in Table 1; columns are labelled according to the treatment (P: pristine, C: crumpled, D: dirty, C + D: crumpled and dirty).

## 2.1.2 Plastic films, soil backgrounds, and experimental workflow

The measurements were performed in case of eight different films (Table 1), where double-sided black and white films were used on both sides and virtually increased the number of films to 10. Each plastic film was analysed in four different conditions: pristine, crumpled, dirty, and crumpled and dirty. Crumpling was obtained by manually rubbing the films to obtain a homogenous and randomized crease pattern (Fig. 1a), while dirtiness was obtained by rolling the films into a dry soil volume, thus creating a layer of soil particles attached to the film (Fig. 1a). A field soil collected on arable land in the Belgian loam belt, classified as silt loam according to USDA classification, and a standard soil (LUFA 2.2), classified as sandy loam according to USDA classification, were used for the


experiment. Both soils were sieved down to 250 µm, to remove coarser organic and inorganic matter, and around 40 g of soil were placed into 87 mm petri dishes. The plastic films were cut to fit into the petri dishes, then the four treatments were applied, and finally, the films were placed on the soil surface in the petri dishes (Fig. 1). Additionally, some petri dishes were left with soil only and used as a reference spectrum of the soil backgrounds (Fig. 1). The use of black nitrile rubber O-rings (Fig. 1) with 85 mm of outer diameter and 75 mm of inner diameter was necessary to fix the films in the petri dishes and prevent wind from blowing them away.

The measurements were performed on a table covered with a black plastic film, and the samples were placed on a black painted wood board (Fig. 1) to minimise background reflectance. The petri dishes and the board were marked to allow for positioning the samples always at the same angle, minimising the influence of sample displacement on optical phenomena. The measurements were carried out in batches of a maximum of 25 samples (Fig. 1). First, hyperspectral measurements were performed on individual samples (Fig. 1a). At the end of the batch, one multispectral image was acquired over all the samples in the batch (Fig. 1b) Overall, 82 samples were analysed, i.e., (10 films x 4 treatments x 2 soil backgrounds) + 2 soils. For each sample, five replicate measurements were carried out throughout the day.

Table 1 - Specifications of the plastic films used for the experiment

| Name | Thickness<br>(µm) | Application mode       | Use                     | Duration use          | Colour      |
|------|-------------------|------------------------|-------------------------|-----------------------|-------------|
| B_1  | 18 ± 2            | On the soil<br>surface | vegetables<br>or fruits | one growing<br>season | black       |
| B_2  | 18 ± 1            |                        |                         |                       |             |
| BIO  | 15 ± 2            |                        |                         |                       |             |
| BW_S | 99 ± 3            |                        | asparagus               | multiple              | black/white |
| WB_S |                   |                        |                         |                       |             |
| BW_L | 143 ± 5           |                        |                         |                       |             |
| WB_L |                   |                        |                         |                       |             |
| T_H  | 44 ± 2            |                        |                         | growing               |             |
|      | 157 ± 1           |                        | vegetables              | seasons               | 4           |
| T_S  | 13/ ± 1           | large tunnel film      | or fruits               |                       | transparent |
| T_L  | 185 ± 6           | for greenhouses        |                         |                       |             |





# 2.1.3 Data processing and analysis

Hyperspectral data were processed in R through the package prospectr (Stevens and Ramirez-Lopez, 2025). First, noisy data at wavelengths below 400 nm and above 2400 nm were removed. Then, a Savitzky–Golay smoothing (Savitzky and Golay, 1964) was applied with a second order polynomial fit and a window size of 11 nm. Finally, water vapour absorption regions, between 1350 and 1450 nm and between 1800 and 1950 nm, were removed. Two spectra, corresponding to one replicate of pristine BW\_S on sandy–loam and one replicate of pristine BW\_L on sandy–loam (Table 1), were removed by visual assessment because of excessive noise, probably due to rapidly changing light conditions.

The multispectral images were pre-processed for radiometric calibration, vignetting correction, black level compensation and conversion to reflectance, following the procedure given by the original manufacturer (Micasense) and using the script provided by Crucil (2021). Following, mean reflectance values were extracted over circular areas of around 5 cm in diameter, centred in each petri dish. This was done to relate the reflectance of an imaging sensor to a non-imaging sensor having a field of view with 4.9 cm of diameter.

Mean values and standard deviation of the five replicates were calculated for each sample (i) and for each wavelength ( $\lambda$ ). Mean spectra are presented for pristine films only. After calculating the average mean ( $\bar{\mu}_i$ ) and the average standard deviation ( $\bar{\sigma}_i$ ) over the whole spectrum, the following coefficient of variation (CV) was used to describe the variation of measurements between replicates of the same sample:

$$CV_i = \frac{\overline{\sigma}_i}{\overline{\mu}_i} \tag{1}$$

To compare the variability across specific groups of film colours and treatments, further aggregated means of the coefficient of variation were also calculated.

To further compare the treatments, the difference in mean reflectance  $(\bar{\rho})$  between the pristine and the treated film were calculated. For each treatment and for each wavelength, the difference  $(\Delta_{\rho})$  is expressed as:

$$\Delta_{\rho_{treatment,\lambda}} = \bar{\rho}_{pristine,\lambda} - \bar{\rho}_{treatment,\lambda} \tag{2}$$

As similarities between spectra of the same film colour groups were observed,  $\Delta_{\rho}$  were aggregated for black, white, and transparent films through mean values.

Lastly, the following three plastic indexes were calculated for hyperspectral data only:

$$HI\_1215 = (1216 - 1197) \frac{\rho_{1235} - \rho_{1197}}{1235 - 1197} + \rho_{1197} - \rho_{1216} \qquad \text{(Garaba and Dierssen, 2018)}$$

$$HI_{-}1732 = (1729 - 1705) \frac{\rho_{1741} - \rho_{1705}}{1741 - 1705} + \rho_{1705} - \rho_{1729}$$
 (Kühn et al., 2004)

$$ND_{-}1715 = \frac{\bar{\rho}_{1590to1630} - \bar{\rho}_{1695to1735}}{\bar{\rho}_{1590to1630} + \bar{\rho}_{1695to1735}}$$
 (Castagna et al., 2023)

Where  $\rho_x$  is the reflectance at the wavelength x, and  $\bar{\rho}_{x\,to\,y}$  is the average reflectance from wavelength x to wavelength y. HI (hydrocarbon indexes) are narrow-band indexes involving three bands, where the middle band represents the wavelength at the absorption peak, and the other two are approximately located before and after the absorption region. ND\_1715 captures the same absorption region represented by HI\_1732, but it is a broadband index calculated through a normalised difference between the reflectance before the absorption region and the reflectance at the absorption region. ND\_1715 was calculated to assess better the potential of a multispectral imaging sensor on a moving platform, which may require broader bandwidths to increase signal-to-noise ratio. To differentiate index values of plastic films from other possible targets (i.e. soil and crop residues), we have extracted index values of soils and crop residues from open spectral libraries (Hively, 2021; Kokaly, 2017).

### 70 2.2 Remote sensing



## 2.2.1 Test site and equipment

An agricultural field located in the province of Córdoba, Spain (37°59'42.30" N, 4°27'40.41" W) was selected to test UAV capabilities in detecting plastic film residues (Fig. 2). The field is located next to the Guadalquivir River. The typical soil at this location is a calcaric Luvisol soil according to FAO classification (Junta De Andalucía, 2005). Despite management practices not including the use of plastic directly on the field, a few plastic items were found on the field, probably resulting from littering, plastic contamination from neighboring fields, or plastic parts of machines used on the field.

UAV images were acquired with a Phantom 4 pro (SZ DJI Technology Co., Ltd.) equipped with a downwelling light sensor and the same multispectral camera used in the proximal sensing experiment (Micasense RedEdge-MX RX02; Fig. 1b; AgEagle Aerial Systems, Inc.). The images were acquired on 28 September 2022, between 11:30 and 12:00, with clear sky conditions. Images of a reflectance panel were acquired before the flight and used for the radiometric calibration of the images in a later stage. The flight height was set to 7 m, providing images with a spatial resolution of around 0.5 cm/pixel. Additionally, two Reach RS2 RTK GNSS receivers (Emlid Tech Kft.) were used for placing the plastic films on the field and acquiring their coordinates.




Figure 2 – Location of the study area, distribution of plastic films in the study area (a), and field picture (b). The geometry of the study area was chosen to maximise the number of plastic films, that are represented by coloured dots (a). On the lower-left corner, the elevation was calculated relatively to the lowest point on the field to highlight tillage patterns.

## 190 2.2.2 Plastic films and experimental workflow

Three of the films used in the proximal sensing experiment were used for the remote sensing experiment. With reference to Table 1, the plastic films used were: B\_1, WB\_S, and T\_H. Films were selected to have one black, one white, and one transparent film. The films were cut into 5 cm x 10 cm pieces, and two small holes on the long edges of the films were made to facilitate the penetration of metal spikes that were used to fix the plastic films into the soil. The plastic films were placed on the field using the GNSS receivers, following a random distribution of points previously generated with a GIS (Fig. 2a). After fixing the films into the soil, the metal spikes were covered with soil to avoid their influence on film reflectance. This left a film surface of around 5 cm x 5 cm uncovered and detectable. In total, 21 black films, 19 transparent films, and 13 white films were placed on the field (Fig. 2a). The experiment was originally designed to place a higher number of films, equally distributed among film colours, on a larger area. However, it was not possible to acquire images of the entire area, and the study area was restricted to that shown in Fig. 2. This resulted in an uneven distribution of films among the different colours.

# 2.2.3 Data processing and analysis

Pix4D mapper (Pix4D SA) was used to create orthophoto mosaics through a structure-from-motion algorithm. The same software was used to perform the conversion to reflectance measurements, using data from the downwelling light sensor and the reflectance panel. Reflectance of saturated pixels, acquired as no data by the multispectral camera, was set to the maximum reflectance of 1. The outputs of the pre-processing – five



orthophoto mosaics, each representing one of the five bands of the multispectral camera – were used for further analyses in ArcGIS Pro (Environmental Systems Research Institute, Inc.).

- The centre of each plastic film was found on the images and labelled according to film colour. Then, 312 random points were generated and labelled by visual interpretation. All the points were labelled as 'soil', and the subclasses 'soil', 'shadow', and 'other' mainly representing vegetation residues or unidentified classes were created to allow a better interpretation of the results. Together with the plastic films, these represented the ground observations, which were used for training and testing the classification with a 5-fold cross validation. A buffer of 1 cm was applied on training points to increase training features. The classification was performed through a random forest (Breiman, 2001), with the maximum number of trees set to 50 and the maximum depth of each tree set to 30. After classification, the images were post-processed using a majority filter on 4-connected pixels, increasing the minimum mapping unit to pixel neighbourhoods but minimizing isolated pixel artifacts. Four different datasets were created to compare the results of different spectral and spatial resolutions:
- 1. Multispectral 0.5 cm: It contains the original five multispectral bands, plus the following vegetation index:

  NDVI (Normalized Difference Vegetation Index) =  $\frac{NIR-Red}{NIR+Red}$  (Tucker, 1979)
  - 2. Multispectral 1.2 cm: It contains the same six bands as the multispectral 0.5 cm dataset. The spatial resolution was reduced to 1.2 cm by resampling the raster through nearest neighbour. The resolution was chosen to simulate the resolution obtained from the same multispectral camera at 20 m of flight altitude. This flight altitude is needed to acquire RGB data at 0.5 cm of spatial resolution, following the specifications of the RGB camera mounted on a Phantom 4 Pro.
  - 3. RGB 0.5 cm: It contains the three original visible bands (i.e., red, green, blue).
  - 4. RGB+indexes 0.5 cm: In addition to the three original bands, three band ratios were calculated as follows:

$$\frac{Red}{Green + Blue}$$
;  $\frac{Green}{Red + Blue}$ ;  $\frac{Blue}{Red + Green}$  (3.1; 3.2; 3.3)

This was done to ensure that at least one RGB dataset had the same number of bands as the multispectral datasets, hence providing the same number of input features into the random forest algorithm.

The detection of the plastic films placed on the field was evaluated against the ground observations by calculating the producer accuracy. The presence of false positives instead (i.e., pixels classified as plastic but not covered by plastic) was checked by calculating the expected number of pixels covered by plastic, weighted by the producer accuracy, and comparing them with the actual number of pixels classified as plastic. For each film, the expected number plastic covered pixels ( $N_{pixels}$ ) was calculated as follows:

$$N_{pixels} = N_{films} \times \frac{A_{film}}{A_{pixel}} \times PA \tag{4}$$

Where  $N_{films}$  is the number of films placed on the field,  $A_{film}$  is the area of a single film,  $A_{pixel}$  is the area of a single pixel, and PA is the producer accuracy of the classification.

## 3. Results




## 3.1 Spectra of pristine films and plastic indexes

On both soils, the spectra of the films are divided into three groups (Fig. 3): black (B\_1, B\_2, BIO, BW\_S, BW\_L), white (WB S and WB L), and transparent (T H, T S, T L). White and black film reflectance is not particularly influenced by the background soil (Fig. 3). In contrast, transparent film reflectance changes according to the soil, with thinner films being closer to the soil spectra than thicker films (Fig. 3). Overall, the reflectance of pristine plastic films showed a high coefficient of variation of 40 % on average, while both soils had a similar coefficient of variation of around 4 %. The reflectance of black films is low across the entire spectrum, and the coefficient of variation is 71 % on average, ranging from 10 % for the B 2 on silt loam to 135 % for the BW S on silt loam. White films have the highest mean reflectance and the lowest coefficient of variation, ranging from 6 % to 17 %. The two different white films present a similar spectral shape, being extremely reflective in the visible spectra, and decreasingly reflective through the SWIR, while the thicker film had higher reflectance across the whole spectra on both soils (Fig. 3). The transparent films generally follow the spectrum of the soil background, but the characteristic soil-shaped spectrum is interrupted by absorption peaks in the SWIR, which are shown by white films at the same wavelengths (Fig. 3). Specifically, the absorption peaks were found around the following wavelengths: 1215 nm, 1730 nm, 1765 nm, 2312 nm, and 2352 nm. As a result, HI 1215 and HI 1732 allow distinguishing white and transparent films, in both pristine and treated conditions, from soils and mixtures, and from crop residues (Fig. 4a,b), while black films do not show any relevant value. While both indices guarantee a complete separability between plastic and non-plastic, HI 1732 values are one order of magnitude greater than HI 1215. ND 1715 has values similar to HI 1732, but the broader spectral interval results in outliers for soils and mixtures, represented by rare minerals (e.g., Erionite, Xenotime) overlaying with ND 1715 values of plastic (Fig. 4c). For both transparent and white films, the indexes generally have higher values for films with a higher thickness (Fig. 4).

Figure 3 – Spectra of pristine plastic films and of soils used as background. On the left side, spectra acquired on silt loam; on the right side, spectra acquired on sandy loam. Film colours are represented by different colours, and variation within film colours are represented by line shapes. Abbreviation of film types refer to Table 1.

Figure 4 – Boxplots of plastic index values for plastic films, soils, and crop residues. Values for all black films (i.e., B\_1, B\_2, BW\_S, BW\_L, BIO) were aggregated, as they did not represent any relevant value. Abbreviations of film types refer to Table 1.

# 3.2 Reflectance of plastic film as residues on soil

Compared to pristine films, the treatments decreased the reflectance variability between replicate of hyperspectral measurements. The average coefficient of variation was reduced to 30 % for crumpled films, 9 %

for crumpled and dirty films, and 9 % for dirty films. Overall, crumpling did not influence plastic film reflectance, as the changes in reflectance are within the range of the coefficient of variation for all the films (Fig. 5). On the contrary, the presence of soil on the film surface influenced plastic film reflectance (Fig. 5). This is particularly evident in the case of white films, where the reflectance of pristine films is substantially higher compared to dirty films in the visible spectra, while the differences between the spectra are reduced for higher wavelengths (Fig. 5).

Black film reflectance also changes when covered by soil, and the differences between spectra increase for higher wavelengths, as soil reflectance increases (Fig. 5). Transparent films are the ones affected the least by treatments, with some changes induced by the presence of soil in the case of sandy loam only (Fig. 5). All these differences were consistently observed for the multispectral measurements in proximal sensing, despite generally higher coefficients of variation (e.g., 45 % for pristine films and 15 % for soils, on average).

Figure 5 – Mean difference in reflectance between pristine films and treatments, presented for two different soils used as background in the proximal sensing experiment. Different data acquisition modes are represented by shapes, while different film colours are represented by the colour scale.

## 3.3 Plastic film detection from UAV


The spectral separability between films and soil observed in proximal sensing is confirmed by the multispectral acquisitions from the UAV (Fig. 6). White films have the highest spectral separability from non-plastic classes,




especially in the bands of the visible spectra, while the reflectance slightly overlays with other non-plastic elements when approaching the infrared (Fig. 6). Transparent film reflectance largely overlays with the reflectance of soil and other non-plastic elements, while black films are quite distinct from soils, especially in the infrared bands, where soil reflectance increases (Fig. 6). Overall, all the 0.5 cm datasets had similar performances in detecting the plastic films placed on the field (Fig. 7a), while the multispectral 1.2 cm dataset had higher omission errors. The difference between the 1.2 cm dataset and the 0.5 cm datasets mainly consists in the missed detection of transparent films (Fig. 7a). White films placed on the field were detected with all four datasets in almost every fold, while black films had a generally higher and more fluctuating omission error (Fig. 7a).

Despite the good performances in detecting the plastic films placed on the field, all the datasets overestimated the presence of plastic on the field (Fig. 7b). Overall, plastic overestimation ranges from an average of 21 times for multispectral 1.2 cm dataset to 44 times for RGB 0.5 cm dataset. Black films encountered the highest overestimation compared to the other films, with RGB datasets having a higher overestimation compared to multispectral datasets (Fig. 7b). After black films, transparent films showed the second highest overestimation, finally followed by white films (Fig. 7b). The overestimation was generally related to the confusion of black films with shadows, white films with highly reflective pixels, and transparent films with soil (Fig. 8).

Figure 6 – Reflectance of plastic films divided by colour (i.e., black, transparent, white), compared to soil reflectance, shadow, and other non-plastic objects found on the experimental site.

Figure 7 – On the left side, accuracy (i.e., producer accuracy) of plastic detection with four different datasets (a), separated by film and aggregated (i.e., 'all'). Error bars represent the standard deviation of the producer accuracy within the 5-fold cross validation; the numbers below the bars (N) represent the average number of points used for validation. On the right side, factor of plastic overestimation of four different datasets, separated by film and aggregated (b). Error bars represent the standard deviation of the overestimation within the 5-fold cross validation.

Figure 8 – Detail of the true colour images (a–c) and associated classification results (d–f) obtained with the multispectral 0.5 cm dataset. On images (a–c), the exact location of the plastic films is highlighted. On image (a), the example of a highly reflective soil region is highlighted.







#### 4. Discussion

## 4.1 Spectral reflectance of plastic film residues

A spectral library of white, black, and transparent films was built, nearly covering all plastic film colours used in agriculture. Within white and transparent films, differences related to thickness and transparency of the films mainly affected the mean reflectance, while the resulting spectral shapes were highly comparable (Fig. 3). Differences between black films can hardly be observed, as spectra are flat and the reflectance is low, producing a low signal-to-noise ratio and high coefficient of variations. The mean reflectance of white and transparent films showed much higher variability across replicate measurements compared to soils, resulting in higher coefficients of variation. This is related to both the experimental design and the optical properties of the films. In fact, the measurements were acquired around 4 hours before and 4 hours after solar noon, with solar zenith angle ranging from 32° to 59°, and solar azimuth angle ranging from 103° to 262° (Suncalc.Org). Moreover, the plastic films have a smooth surface, which leads away from the assumption of Lambertian reflector - typically adopted in remote sensing - and causes specular reflection to take over diffuse reflection phenomena (Goddijn-Murphy and Dufaur, 2018; Goddijn-Murphy et al., 2017). This is particularly evident in the case of pristine films, which had the highest coefficient of variation, while the presence of soil on the film surface increases the roughness of the films, favouring diffuse reflection and decreasing the variability of the reflectance between replicates. The implications for plastic residue detection are mostly positive, as particle deposition is expected on land, while specular reflection may be a limiting factor in other applications, such as detection of clean plastic films with satellite data. Despite a decrease in the coefficient of variation, we did not find any relevant influence of crumpling on film reflectance, while soil was the main driver of changes in reflectance between pristine and treated films (Fig. 5). Changes in plastic film reflectance were particularly evident for white films in the visible spectra, where the distance between the spectra of plastic films and soil is the highest (Fig. 3, Fig. 5). The presence of soil did not have a particular influence on plastic absorption peaks, enabling plastic identification through spectral indexes, also on soil-covered plastic films (Fig. 4). However, as the amount of soil on the film increases, a progressive deterioration of plastic spectra is expected, until reaching a soil cover at which plastic films will not be detectable anymore. Identifying this threshold of soil cover will be helpful in delineating the boundaries of remote sensing use for detecting plastic film residues on cropland. We suggest using indoor spectroscopy to that aim, exploiting a higher signal-to-noise ratio and better control on the amount of soil placed on the films compared to outdoor measurements, where wind can remove soil particles from the film surface. We also advise against using a contact probe in indoor spectroscopy, as we experienced plastic film surface melting in contact with the light source.







# 4.2 Use of multispectral and true colour UAV images for plastic film detection

Film spectra acquired from UAV were highly comparable with the proximal sensing acquisitions. White films had very high reflectance for shorter wavelengths and decreasing reflectance for higher wavelengths, transparent films had reflectance similar to soil, and black films had a very low reflectance. It must be accounted for, though, that the reflectance of white films was highly influenced by assigning the maximum reflectance of 1 to no data values, corresponding to saturated pixels. This particularly influenced the blue and green bands, where the number of saturated pixels was more than half, while the red and red edge bands had around 20 % and 10 % of saturated pixels, respectively, and the NIR had no saturated pixels.

Good performances were achieved in detecting true positives of plastic films (Fig. 7a). Despite the presence of other plastic residues on the field, plastic films were overestimated compared to the expected number of plastic-covered pixels (Fig. 7b). Specifically, black films had the highest overestimation, which was due to the presence of shadows on the field (Fig. 8a). In our study area, recent tillage practices induced soil clod formation and emphasized the impact of shadows (Fig. 2). However, surface microtopography is always expected on arable land, and we identify shadows as the main limiting factor in black film detection, similarly to previous works dealing with black plastic detection (lordache et al., 2022; Shan et al., 2018). White films were slightly overestimated in correspondence of highly reflective soil regions (Fig. 8a). As with shadows, the topography of our study area might have emphasized the issue, creating soil surface angles that result in specular reflection. While techniques like brightness thresholding could improve the results of the classification (lordache et al., 2022), their implementation and effectiveness may be site-specific, and a few measures could be adopted during data collection to address the challenges related to the illumination geometry. Possible solutions are multi-temporal flights, high-resolution DSM (Digital Surface Model), or flying with a homogenous cloud cover and the highest solar elevation angle possible.

While transparent films had a lower overestimation compared to black films (Fig. 7b), the source of error is related to a strong spectral overlay with soil in the visible and near infrared (Fig. 3, Fig. 6), rather than a structural flaw of the survey. In this case, the only solution to substantially improve plastic detection may be limited to having sensors with bands located in the SWIR. The additional bands on the multispectral camera did not show a substantial improvement in transparent plastic detection, especially when compared with RGB+indexes 0.5 cm dataset (Fig. 7). For black films, the datasets showed similar producer accuracies, but the multispectral datasets showed a lower overestimation compared to the RGB datasets (Fig. 7). However, the difference in spatial resolution must be considered, as coarser pixel size negatively affects both the areal estimates and the minimum mapping unit. At the same time, for equal pixel size, RGB cameras allow for increasing flight height, consequently reducing flight time and increasing area coverage. Overall, the results of our study suggest that the increased








spatial resolution of an RGB camera should be favoured over the higher spectral resolution of multispectral cameras.

Lastly, it needs to be highlighted that using more sophisticated algorithms like deep learning and image segmentation techniques would likely increase detection accuracy. However, our experimental setup might have introduced bias when using textural features (e.g., fixed size and shape of plastic films, placement of plastic films under soil clods) and finding the best image classification algorithm goes beyond the scope of our study. Instead, a random forest allowed us to relate the results to spectral features, and to define which sensors and survey workflow are best suited for plastic films detection on soil.

## 4.3 A new generation of sensors for monitoring plastic on soil

We have shown potential workflows for detecting plastic film residues with RGB or multispectral cameras. However, the accuracy of these approaches will be site and film specific. A clear sandy soil may represent the optimal site for detecting black films, providing a good contrast between the film and the background, and reducing the complexity of the topography by limiting the formation of soil clods. At the same time, such soils are more likely to induce sensor saturation and a lower contrast with clear films, making the detection of white films harder. On the contrary, a dark clay soil may provide a good environment for white film detection, while black film detection would be limited by a decreased contrast and eventually hindered by the presence of soil clods and direct sunlight. In general, plastics do not show any unique feature in the visible and near infrared (Fig. 3), and their spectral separation with currently available multispectral broadband sensors is colour-driven.

While the use of plastic absorption bands in the aquatic environment may be limited by water absorption features (Knaeps et al., 2021; Moshtaghi et al., 2021; Garaba and Dierssen, 2018), their use is extremely promising for soils, with plastic indexes providing an unambiguous identification of the plastic residue (Fig. 4). Even the broadband index ND\_1715 resulted very effective in identifying plastic films from other elements that could possibly be found on cropland – except for some rare minerals that will not be found on arable land (Fig. 4c) – encouraging the development of multispectral cameras tailored to plastic detection. The availability of broad bands related to SWIR absorption features on miniaturised sensors, together with high-resolution visible bands, would reduce costs, data volume, and complexity of currently available hyperspectral sensors, representing a milestone for monitoring any non-black plastic residue on land.

Additionally, according to Castagna et al. (2023), remote sensing offers the potential to distinguish at least three plastic types, exploiting the different locations of absorption features in the SWIR. PE, PP, and PVC – the most produced plastic types in Europe and the most used polymers in agriculture (Plastics Europe Aisbl, 2024) – have similar absorption features, and their distinction might be challenging (Castagna et al., 2023). PET instead – the fourth most produced plastic type in Europe (Plastics Europe Aisbl, 2024) – has different absorption features, and it can be distinguished from PE, PP, and PVC by using narrow-band indexes (Castagna et al., 2023). On




agricultural land, this could mean distinguishing plastic residues generated by agricultural management from plastic residues generated by littering (e.g., PET bottles). Building spectral libraries and characterising the absorption peaks of the most common plastic residues is key to direct future efforts in developing new sensors, which we identify as a necessary to monitor soil plastic contamination. The introduction of legal limits on plastic concentration in soil may not be far off, as initial regulatory examples begin to emerge (Meixner et al., 2020). This will likely drive demand for faster and standardized monitoring and increase the appeal of a sensor for on-soil plastic detection. Moreover, as macroplastic residues can fragment into microplastic (Yang et al., 2022; Song et al., 2017; Julienne et al., 2019), reliable macroplastic monitoring will also support better microplastic assessments.

## 5. Conclusion

The introduction of miniaturised multispectral imaging sensors has boosted the use of UAV remote sensing. Compared to RGB cameras, multispectral sensors provide access to a few additional bands, typically at the cost of reduced spatial resolution and increased complexity of use. The additional bands available on commercial multispectral sensors are mostly designed for vegetation monitoring, while the detection of plastic residues still relies on the identification of their colour. Within current technologies, our results support using RGB cameras for monitoring plastic film residues on agricultural soils with UAVs, favouring spatial resolution over spectral resolution. However, the accuracy of these techniques will be highly dependent on the type of soil and on the colour of the residue.

The use of plastic absorption bands provides an unambiguous identification of the plastic residue, especially on soils. Currently, these bands are available only on expensive and complex sensors, hampering their use for monitoring large scales, at which remote sensing is most needed. Plastics are ubiquitous, persistent, and bioaccumulative contaminant that require monitoring plans, and the development of miniaturised sensors targeting plastic detection may represent a pivotal moment for assessing the risk of plastic contamination on land.

We expect this work to promote further the exploration of remote sensing potential for monitoring soil plastic contamination and to encourage building and sharing spectral libraries of plastic residues commonly found on soils.

## **Credit author statement**

All authors: writing – review and editing, conceptualization, methodology, resources. A.F., F.W., P.F.: visualization. F.W., P.F.: supervision, project administration, funding acquisition. A.F.: validation, investigation, software, formal analysis, data curation, writing – original draft.




# Acknowledgements

This project has received funding from the European Union's Horizon 2020 research and innovation programme under the Marie Skłodowska–Curie grant agreement No 955334. We thank the Institute of Sustainable Agriculture (IAS–CSIC, Córdoba) for the access provided to the research infrastructures. We thank Marco Bravin and Ana Carolina Cugler Moreira for their contribution to the data collection. We thank the members of the Water and Soil Resource Research group of the University of Augsburg for their constant and valuable feedback.

# **Competing interest**

At least one of the (co-)authors is a member of the editorial board of SOIL.

## 455 Data availability

The spectral libraries used in this research are available at: https://doi.org/10.5281/zenodo.14336253

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
