# Peer review of "Plastic film residues on cropland: monitoring soil contamination through optical remote sensing"

_EGUsphere, 2025_

## Author Comment (AC2)

**Reply to general comments - Referee #2**

**Plastic film residues on cropland: monitoring soil contamination through optical remote sensing**

Alessandro Fabrizi, Peter Fiener, Kristof Van Oost, Florian Wilken

We thank the reviewer for acknowledging the study and for providing constructive feedback. We are glad that the reviewer shared their expertise to identify critical points and improve the quality and reproducibility of the study. In this first part of the review process, we would like to address the general comments and the comments we believe are more critical towards further steps in the publication process.

Please find below our detailed answers (in italics):

**Comments**

1. First, the field dataset is relatively small, raising questions as to how robust and reliable the results are.
   More specifically, the study uses a small number of plastic film samples in the field study, especially considering that they were used to train random forest models. Twenty-one black films, 19 transparent films, and 13 white films were placed on the field, and for validation purposes, only 4.2, 3.8 and 2.6 number of points were used for validation for the black, transparent and white plastic films (according to Figure 7), respectively. The concern is that the training and validation of the classification models are not reliable enough.

*We agree that a higher number of plastic film samples would have been beneficial for interpreting the results. However, the field experiment aimed to simulate a UAV survey on an agricultural field using plastic films, and the surface occupation of plastic films relative to the overall area of the field was designed to replicate a realistic scenario. We believe that this issue will recur with plastic film residue mapping on agricultural land. Heavily polluted agricultural fields are rarely documented, especially in Europe, and finding a high number of plastic film samples bigger than 5 cm for training and validation will be the exception.*

*In the revised manuscript, we will include a sentence to justify the choice of the low number of plastic film samples and discuss limitations.*

2. In the methods section, information is lacking on the soil used in this study, as little or no chemical or physical data is provided (e.g. organic matter, pH, etc.). Soil properties can greatly affect spectral results as thus are important in this study.

*Thank you for the feedback. One of the soils used in the proximal sensing experiment is a standard soil provided by a German company, and we will provide all relevant information from laboratory analyses. The other soil used in proximal sensing and the soil from the field experiment could be provided with regionally specific values, but no laboratory analyses were performed.*

3. The description of the plastics used in this study it is not detailed enough. Which type of plastics was used, beyond the plastic color? This information is vital, especially as one of the aims of the study was to 'build spectral libraries for the most common plastic films used in agriculture' (line 180).

*We agree that the information is vital and is missing. The plastic used was LDPE, obtained from plastic film manufacturing companies. We will include the information in the revised manuscript and clarify the details on plastic films provided in Table 1 based on the reviewer's feedback.*

4. Additionally, some of the methods are not described sufficiently- for example the process of calculating the producers' accuracy in the field study. How many points were used for validation? How were validation points created? This is especially important with the small dataset that was used in this study.

*The producer accuracy was calculated as the ratio between the number of plastic films correctly detected and the number of plastic films placed in the field. The producer accuracies provided in Figure 7 represent the mean and standard deviation of the producer accuracy calculated across the 5-folds used in the study. In each fold, 80% of the ground observations were used for training, and 20% were used for validation, ensuring that the observations used for validation were different at each fold. The validation points were created by extracting the centroid of the plastic films placed on the field.*
*We will add this information to the revised manuscript.*

5. In the results section, important results are not displayed, such as the spectra of the crumpled, dirty, and crumpled + dirty plastics films.

*We agree that the spectra of the treated films could add value to the scientific community, and we will include them in the supplementary material of the revised manuscript. However, we believe that Figure 5 and the numbers provided on the coefficients of variation in section 3.2 already highlight the changes occurring from pristine films to crumpled, dirty, and crumpled + dirty films, which is one of the main objectives of the experiment.*

*Regarding Figure 4, the values of the indices include pristine films, as well as crumpled, dirty, and crumpled + dirty films. Despite showing the treatments in separate figures could increase the detail of the manuscript, we believe that the separability of plastic with the indices can already be appreciated in the figure. Hence, we would like to present the treatments in a single figure to limit the number of figures and streamline the reader's attention. Instead, we will add details to the caption of Figure 4 in the revised manuscript.*

6. Images or results from the multispectral images acquired in the controlled outdoor experiment and of the field in the field scale study are not shown as would be expected.

*Thank you for the feedback. We will include an RGB composite image of the field-scale study in the revised manuscript. However, we are not quite sure about how the reviewer would like to include images of the controlled outdoor experiment. We believe that Figure 1 already shows the setup of the experiment and the differences between the plastic film samples. Moreover, several multispectral images were acquired, as we had 82 samples and 5 replicate measurements for each sample.*

7. Line 113-114: how was it ensured that all the plastic films were equally 'dirty'? was the entire plastic film covered by soil? From Figure 1, it seems like there is a relatively thick layer of soil that mostly obscures some of the plastic films, but not others. It is hard to understand if that was actually so from the description. which soil was used to make the films dirty? the choice of soil could greatly affect the spectra (clayey vs sandy soils for example).

*The plastic films were not equally dirty. Instead, the methodology to make the films dirty was the same. As described at lines 113-114, 'dirtiness was obtained by rolling the films into a dry soil volume, thus creating a layer of soil particles attached to the film'. We agree that differences between films are visible in Figure 1, and the amount of soil on the films was not equal. However, we believe that this represents field conditions. Different plastic films exhibit varying properties, such as surface roughness or surface resistivity, which influence soil*

*particle retention on their surface. Moreover, plastic particles will be more or less visible depending on the contrast with the background. Finally, both soils were used to make the films dirty, and the results are shown separately in Figure 5.*

*We apologise for the missing information, and we will add these details to the revised manuscript in the methodology.*

8. Lines 185-190: How was elevation calculated? From the UAV images? If so, what was the flight overlap and what method was used to calculate elevation?

*The elevation was calculated from the digital surface model obtained from UAV image processing in Pix4D. The elevation was calculated relatively to the lowest point on the field. Flight overlap was set to 70%. However, this may have decreased during the flight, as flight altitude was observed to be variable during image processing. We are not quite sure whether this was related to a real decrease in flight altitude or to an incorrect data acquisition from the UAV.*

*We will add these details to the revised manuscript, but we plan to substitute the image of Figure 2a with the RGB composite. Both reviewers requested the inclusion of the RGB composite, and we would avoid increasing the number of figures.*

9. Lines: 204-209: were Ground Control Points (GCPs) placed in the field for generating the orthomosaics?

*One GCP was on the part of the field used as the study area in this work. The GCP was part of five GCPs placed on a larger portion of the field, initially designated as our study area. However, the GCP was not used for generating the orthomosaics. We are aware that this does not guarantee correct georeferencing of the images and may have caused some band misalignment, which is visible in band composites through halo artefacts. However, we believe that a correct georeferencing of the images may be more relevant in applications such as change detection. Moreover, our field study does not aim at providing a benchmark for plastic film detection. Our results are used to define the ideal sensor for plastic film detection by comparing visible spectra with non-visible spectra on a multispectral camera. The bands are compared as part of the same camera during the same survey.*

*We will add more details on the UAV survey and provide a more in-depth discussion of survey flaws related to the study's goal in the revised manuscript.*